# Properties and Applications of Metal Phosphates and Pyrophosphates as Proton Conductors

**DOI:** 10.3390/ma15041292

**Published:** 2022-02-09

**Authors:** Rosario M. P. Colodrero, Pascual Olivera-Pastor, Aurelio Cabeza, Montse Bazaga-García

**Affiliations:** Departamento de Química Inorgánica, Cristalografía y Mineralogía, Facultad de Ciencias, Universidad de Málaga, Campus Teatinos s/n, 29071 Málaga, Spain; colodrero@uma.es (R.M.P.C.); poliverap@uma.es (P.O.-P.); aurelio@uma.es (A.C.)

**Keywords:** metal phosphate, proton conductivity, super protonic conductors, fuel cells, H-bond network, proton carriers

## Abstract

We review the progress in metal phosphate structural chemistry focused on proton conductivity properties and applications. Attention is paid to structure–property relationships, which ultimately determine the potential use of metal phosphates and derivatives in devices relying on proton conduction. The origin of their conducting properties, including both intrinsic and extrinsic conductivity, is rationalized in terms of distinctive structural features and the presence of specific proton carriers or the factors involved in the formation of extended hydrogen-bond networks. To make the exposition of this large class of proton conductor materials more comprehensive, we group/combine metal phosphates by their metal oxidation state, starting with metal (IV) phosphates and pyrophosphates, considering historical rationales and taking into account the accumulated body of knowledge of these compounds. We highlight the main characteristics of super protonic CsH_2_PO_4_, its applicability, as well as the affordance of its composite derivatives. We finish by discussing relevant structure–conducting property correlations for divalent and trivalent metal phosphates. Overall, emphasis is placed on materials exhibiting outstanding properties for applications as electrolyte components or single electrolytes in Polymer Electrolyte Membrane Fuel Cells and Intermediate Temperature Fuel Cells.

## 1. Introduction

Metal phosphates (MPs) comprise an ample class of structurally versatile acidic solids, with outstanding performances in a wide variety of applications, such as catalysts [1,2,3], fuel cells [4,5,6], batteries [7], biomedical [8], etc. Depending on the metal/phosphate combinations and the synthetic methodologies, MP solids can be prepared in a vast diversity of crystalline forms, from 3D open-frameworks, through layered networks, to 1D polymeric structures.

The advantages of these solids are, among others, their low cost and easy preparation, hydrophilic and thermally stability and structural designability to a certain extent. Additionally, their structures are amenable to post-synthesis modifications, including the incorporation of ionic or neutral species that significantly affect their functionality and other important properties, such as the formation of hydrogen bonding networks or insertion of proton carriers.

While tetravalent and monovalent metal phosphates are usually prepared by conventional synthetic routes, the synthesis of many divalent and trivalent metal phosphates requires the presence of organic molecules/ions as structure-directing agents, some of which also act as charge-compensating ions [9]. In any case, the metal phosphates contain acidic phosphate groups (H_2_PO_4_^−^ and HPO_4_^2−^) or they can result from specific post-synthesis acid–base reactions, thus, allowing a mostly intrinsic proton conduction.

Additionally, an extrinsic proton conduction, associated with surface proton transport, can also result from specific chemical modification and/or induced morphological changes, e.g., with formation of nanoplatelets or nanorods particles [10,11]. Water molecules act both in the formation H-bonding networks inside the framework and as proton carriers. Moreover, other guest species, including organic moieties, can be inserted in their structure following different strategies, such as intercalation [12,13,14], template [15,16], and post-synthesis adsorption [17,18,19].

Accordingly, proton transport in MPs results from the interplay between water/guest molecules and the hydrophilic network/framework walls in specific ways. Thus, in metal phosphates, such as CsH_2_PO_4_ [20] or CaHPO_4_·2H_2_O [21] the proton-transfer mechanism involves not only the jump of H^+^ but also the free rotation of the phosphate groups. In other cases, such as for KH_2_PO_4_ the proton-transfer mechanism behavior is by defects [22].

A more complex proton-transfer process arise from the so called “quasi-liquid” state, such as described for H_3_OUO_2_PO_4_·3H_2_O [23,24], or it may stem from various mobile species moving on the surface, each one using different paths at different rates, e.g., in Ce(HPO_4_)_2_·nH_2_O [25,26]. The proton transport in the quasi-liquid state is often characterized by a “proton jump” or the Grotthuss mechanism, with low activation energy (E_a_) [21].

Determining proton conduction mechanisms is key to designing proton conductor materials (Figure 1). Thus, if a vehicle-type mechanism is needed, proton-containing counter ions, such as hydronium ions, NH_4_^+^, protonated amine groups, and protonated organic molecules in general, should be incorporated into the structure. Additionally, the immobilization of specific functional groups and the corresponding counter ions onto the framework could be required.

On the other hand, solids exhibiting proton transport by a Grotthuss-type mechanism basically consist of continuous H-bond networks that favour H^+^ conduction with low activation energy (E_a_). A main drawback for the latter materials is that, above certain temperatures, breaking of the H-bond networks can result, accompanied by the release of water molecules, which makes the construction of permanent H-bonding networks with hydrophilic channels or exploring new alternative conductive media necessary [27].

A research area of paramount interest is that relative to fuel-cell (FC) development regarding energy efficiency, environmental issues, and the gradual depletion of conventional fuel reserves [28,29]. The operating temperature regime of these electrochemical devices is determined by the electrolyte. Among the many types of solid electrolytes that have been developed over the past several decades [30], perfluorinated sulfonic acid (PFSA) ionomers, such as Nafion^®^ and Aquivion^®^, are the most widely used in PEMFCs and DMFCs as polymer electrolyte membranes (PEM) for operating temperatures <120 °C.

This is due to superior conductive properties, ranging from 9 × 10^−3^ S·cm^−1^ to 1.2 × 10^−1^ S·cm^−1^, and chemical and mechanical stability [31,32]. This PEM acts as a separator for the reactants and, at the same time, provides pathways for proton transport [33,34]. However, Nafion^®^ also shows some important drawbacks, such as methanol permeability, high costs, and complex steps of preparation, working below 100 °C to maintain the hydration of the membrane and other issues related to catalyst efficiency. All this makes necessary the development of new membrane materials [35,36].

Intermediate Temperature fuel cells (ITFCs), which operate between 120 and 300 °C, circumvent some of these problems, achieve less catalyst poisoning and better handling of the hydration conditions, and enable the use of non-precious or low-loading precious metal catalysts [37,38,39]. Nevertheless, for the case of methanol HT-PEMFCs, which must operate above 250 °C and with high pressure, possible degradation of the electrolytes should be a subject of study [40]. Some MP-based electrolytes gather proton conductivity properties to be used as single electrolytes or main components for ITFC applications. Examples are superprotonic CsH_2_PO_4_ [41] and M(IV)P_2_O_7_-based electrolytes [42].

This review summarizes the state of the art on proton conductive metal phosphates with special attention to the last two decades of developments. A review of publication trends, from the CAS SciFinder^n^ database, is given in Figure 2, which confirms the increasing interest in this topic during this period. In this article, our focus is placed on updating recent advances in proton conductors based on tetravalent metal phosphates/pyrophosphates and superprotonic Cs_2_HPO_4_. In addition, the design and development of new proton carrier systems relying on divalent and trivalent metal phosphates are also revised.

The potential applications of these materials as proton conductor electrolytes for PEMFCs and ITFCs are also discussed. Other high temperature (above 300 °C) conductive metal phosphates, such as rare earth phosphates and metal polyphosphates, ionic conductors (e.g., NASICON), and electrode materials are out the scope of this revision. The review is not intended to be exhaustive, and therefore certain specific reports and/or applications may not be covered.

## 2. Tetravalent Metal Phosphates and Pyrophosphates

### 2.1. Zirconium Phosphates

The prototype of layered metal(IV) phosphates is zirconium hydrogen phosphate, which presents mainly three topologies, known as the forms **α**: Zr(HPO_4_)_2_·H_2_O (α-ZrP), **γ**: Zr(PO_4_)(O_2_P(OH)_2_) 2H_2_O (γ-ZrP), and **λ**: (Zr(PO_4_)XY, X = halides, OH^−^, HSO_4_^−^, and Y = DMTHUS, H_2_O, etc.) (λ-ZrP). α-ZrP is the most thermal and hydrolytically stable phase and, thus, the most studied compound following the pioneering works by Giulio Alberti and Abraham Clearfield groups [43,44,45].

In the structure of α-ZrP, planar Zr atoms are coordinated by the three oxygen atoms of HO-PO_3_^−^ groups, while the P-OH bond points toward the interlayer space giving rise to small cavities, where the lattice water is located. Although this water forms hydrogen bonds with P-OH groups, an extended interlayer hydrogen bond network is absent in this structure [44]. On the other hand, the structure of γ-ZrP consists of a biplanar Zr atomic system connected through bridge PO_4_^3−^ groups, while the externally located H_2_PO_4_^−^ groups complete the octahedral coordination of the Zr atoms.

Due to the higher stability of the α form, proton conductivity studies have been conducted mainly with this phase. Typical values of 10^−6^–10^−4^ S·cm^−1^ have been determined at RT and 90% RH with activation energy (E_a_) ranging between 0.26 (90% RH) and 0.52 eV (5% RH). This conductivity originates from surface transport and is dominated by surface hydration since the long distance between adjacent P-OH groups hinders proton diffusion along the internal layer surface [46].

According to theoretical estimations [47], the interactions between POH and water molecules on the zirconium phosphate surface are relatively large and generate strong local hydrogen-bond networks. The high density of acid groups should produce continuous short O-O distances and generate proton-transfer paths with low activation energy, even under high-temperature and low-humidity conditions.

Studies on microcrystalline α-ZrP revealed that both the morphology and the electric field orientation are important aspects to consider. Thus, pellicular samples of α-ZrP, prepared by intercalation-deintercalation of n-propylamine [48], exhibited a proton conductivity of 10^−4^ S·cm^−1^ at 300 °C when it was measured with the electric field parallel to the pellicles. This conductivity was found to be approximately two orders of magnitude higher than that measured on applying the electric field perpendicular to the pellicles [49].

Several other strategies to increase the proton conductivity of α-ZrP have been developed (Table 1). One of the first approaches was the formation of polyhydrated n-propylamine-intercalated compounds (ZrP·xPrNH_2_·nH_2_O). For a composition with x = 0.8 and n = 5, a proton conductivity of 1.2 × 10^−3^ S·cm^−1^ at 20 °C and 90% RH was determined [50]. Another more sophisticated way of enhancing the proton conductivity of zirconium phosphate was the preparation of mesoporous zirconium phosphate/pyrophosphate, Zr(P_2_O_7_)_0.81_(O_3_POH)_0.38_.

This material, obtained from the thermal decomposition of zirconium phosphite benzenediphosphonate, displayed a high specific surface area (215 m^2^/g) with an enhanced proton conductivity of 1.3 × 10^−3^ S·cm^−1^ at 20 °C and 90% RH [51]. One of the most successful ways of obtaining zirconium phosphate derivatives with outstanding proton conductivity properties consisted in preparing mixed zirconium phosphate/phosphonates, an example being [Zr(O_3_POH)_2−x_(O_3_PC_6_H_4_SO_3_H)_x_], whose proton conductivity was found to increase with *x*, up to a value of 0.07 S·cm^−1^ (x = 1.35 at 100 °C and 90% RH) [52].

The great versatility of zirconium phosphate preparation under numerous experimental conditions, has been exploited to synthesize various novel derivatives, which displayed remarkable proton conductivity properties. Two examples are layered (NH_4_)_2_[ZrF_2_(HPO_4_)_2_] [53] and the 3D open framework zirconium phosphate (NH_4_)_5_[Zr_3_(OH)_3_F_6_(PO_4_)_2_(HPO_4_)] [54]. In both structures, the participation of NH_4_^+^ ions in the formation of extended hydrogen-bond networks (Figure 3) resulted in proton conductivities of 1.45 × 10^−2^ S·cm^−1^ (at 90 °C and 95% RH) and 4.41 × 10^−2^ S·cm^−1^ (at 60 °C and 98% RH), respectively. From the E_a_ values found, 0.19 and 0.33 eV, respectively, a Grotthuss-type proton-transfer mechanism was inferred for these materials (Table 1).

A one-dimensional zirconium phosphate, (NH_4_)_3_Zr(H_2/3_PO_4_)_3_, consisting of anionic [Zr(H_2/3_PO_4_)_3_]_n_^3n−^ chains bonded to charge-compensating NH_4_^+^ ions was reported [55]. In this structure, the phosphates groups are disorderly protonated, while NH_4_^+^ ions occupy ordered positions in between adjacent chains (Figure 4). This arrangement generates H-bonding infinite chains of acid–base pairs (N-H···O-P) that lead to a high proton conductivity in anhydrous state of 1.45 × 10^−3^ S·cm^−1^, at 180 °C (Table 1). This material also showed a remarkable high performance in PEMFC and DMFC under operation conditions.

### 2.2. Zirconium Phosphate Composite Membranes

ZrP has been widely studied as a suitable filler for the preparation of Nafion^®^-composite electrolytes due to its good thermal and chemical stability, proton conductivity, hydrophilic nature, very low toxicity, and stability in hydrogen/oxygen atmospheres. Nafion^®^, a sulfonated tetrafluoroethylene-based fluoropolymer copolymer is the most widely used proton conductor for polymer electrolyte membrane fuel cells (PEMFCs) [56].

Despite its excellent properties, this polymer is limited to operations at low temperatures of <100 °C, due to dehydration, which drastically reduces the proton conductivity. As operating at higher temperatures is desirable for better efficiency, Nafion^®^ composite membranes have been investigated in which fillers of diverse nature were incorporated to the polymer using different approaches.

Various methodologies of preparing Nafion^®^/ZrP membranes have been devised [46,57,58,59,60,61,62,63,64,65]. Among them, treatment of the Zr(IV)-exchanged ionomer with phosphoric acid [59], filler/ionomer co-precipitation [61] and casting of ionomer solution/filler gel mixtures [60,61,62]. In general, improvements in mechanical properties and efficiency [61] were highlighted, although high filler loads beyond 25% led to decreased proton conductivity in certain cases.

A derivation of the above methods consisted of using nanosized ZrP particles-containing ionomer Aquivion^®^ as fillers [63]. The resulting composite membranes showed enhanced elastic modulus and lower proton conductivity as compared to Nafion^®^ itself, although they were high enough for fuel cell application [64] with values ranging between 0.16 and 0.23 S·cm^−1^ at 90% RH [62,65].

Derivatising ZrP with hydrophobic groups resulted in composite membranes mechanically stronger than the pristine Nafion^®^ membrane while displaying high proton conductivities [66,67,68,69,70]. Furthermore, integrating both hydrophobic and hydrophilic ZrP nanofillers showed a synergistic effect with mechanical reinforcement of the composite membrane and better fuel cell performance than that found when single ZrP filler were employed [60]. Other ionomer alternatives to Nafion^®^, such as sulfonated poly(arylene ether) and poly(ether ether ketone) polymers (SPAEK and SPEEK, respectively), polybenzimidazole (PBI), and vinyl-type (PVA and PVP), have been employed as components of polymer electrolyte membranes [71].

The sulfonated poly(fluorenyl ether ketone) (SPFEK) polymer presents acceptable proton conductivity and stability; however, its performance is lesser in comparison to the benchmark Nafion^®^ ionomer. To overcome this handicap, several approaches have been implemented, one of them being hybridization with α-zirconium phosphate. Recently, SPFEK/ZrP-SO_3_H composite membranes have been reported with improved proton conductivity, oxidative stability, methanol cross-over barrier and improved H_2_/O_2_ fuel cell performance [72,73].

ZrP has been also used as a filler in other polymeric matrices, such as poly(vinylidene fluoride) (PVDF) and (sulfonatedpoly(styrene-block-(ethylene-ran-butylene)-block-styrene) (S40). The resulting composite membranes were tested for potential applications in DMFC, with improved performance especially relative to reducing the methanol permeability [72,74]. PA-PBI/Zr(HPO_3_) composite membranes (PA-PBI = phosphoric acid-doped polybenzimidazole) exhibited enhanced proton conductivity as compared to a pristine polymer electrolyte membrane at 200 °C and 5% RH [75].

A different approach is based on the substitution of the polymer membranes for others more eco-friendly, inexpensive, and biodegradable materials. Although poly(vinyl alcohol) (PVA) is a suitable candidate for this purpose, it still requires some enhancements related to its ionic conductivity due to its rigid structure. As an example, Gouda et al. [76] doped iotan carrageenan (IC)/sulfonate PVA (SPVA) membranes with different loads of zirconium phosphates. Enhancement of the membrane properties was attributed to abundant hydrogen bond interactions established between the zirconium phosphate particles and the polymers.

### 2.3. Titanium and Tin(IV) Phosphates

Structurally, titanium(IV) phosphates are more diverse than ZrP, given that structures containing an oxo/hydroxyl ligand or mixed-valence (Ti^III^/Ti^IV^) titanium ions have been reported. In addition to the zirconium phosphate analogues, α- [77] and γ-TiP [78], two more other layered titanium phosphates, TiO(OH)(H_2_PO_4_)·2H_2_O [79] and Ti_2_O_3_(H_2_PO_4_)_2_·2H_2_O are known [80]. Furthermore, the mixed-valence titanium phosphate Ti^III^Ti^IV^(HPO_4_)_4_·C_2_N_2_H_9_·H_2_O contains microporous channels [81], where monoprotonated ethylenediamine and lattice water are accommodated, which favours its transformation into a porous phase, Ti_2_(HPO_4_)_4_, at 600 °C with a 3D structure similar to that of τ-Zr(HPO_4_)_2_ [82].

Impedance spectroscopy experiments combined with dynamics simulations revealed that this compound was a good proton conductor, with a σ value of 1.2 × 10^−3^ S·cm^−1^ (at room temperature and 95% RH) and E_a_ of 0.13 eV (Table 1). From these data, a water-mediated proton transport originating from the H-bond interaction of water molecules with the bridging oxygen of the porous framework was proposed [83].

The crystal structure of two other open-frameworks Ti_2_O(PO_4_)_2_·2H_2_O polymorphs (ρ-TiP and π-TiP) has been reported [84,85]. Fibrous ρ-TiP and π-TiP crystallize in triclinic and monoclinic unit cells, respectively, and are composed of TiO_6_ and TiO_4_(H_2_O)_2_ octahedra bridged through the orthophosphate groups (Figure 5). This type of connectivity creates two types of 1D channels running parallel to the direction of the fibre growth [84,85,86]. ^31^P MAS-NMR spectroscopy studies revealed that π-TiP has capability of adsorbing superficially protonated phosphate species (H_3_PO_4_/H_2_PO_4_^−^/HPO_4_^2−^), which affects the proton conductivity properties as confirmed by AC impedance measurements.

Thus, this solid reached a proton conductivity value of 1.3 × 10^−3^ S·cm^−1^, at 90 °C and 95% RH (Table 1). This behaviour was explained as being due to an extrinsic vehicle-type proton transport mechanism. Additionally, H_3_PO_4_-impregnated π-TiP solid was used as filler for the preparation of composite membranes of Chitosan (CS) matrices exhibiting a proton conductivity of 4.5 × 10^−3^ S·cm^−1^ at 80 °C and 95% RH, which is 1.8-fold higher than that of the pristine CS membrane [86].

Insights in the structure of the isomorphous series of layered α-metal(IV) phosphates [M(IV) = Zr, Ti, Sn) showed appreciable differences in the hydrogen bond interaction of the lattice water with the layers, associated with a variable layer corrugation degree along the series; α-TiP and α-SnP displaying H-bonds stronger than the prototype α-ZrP [87], which, in turn, might have significant implications in making internal surfaces accessible to proton conduction paths.

Nevertheless, studies on structure–property correlations are still lacking, due, in part, to appreciable differences in chemical stability under different conditions, particularly in the case of α-SnP. Recent studies carried out on a nanolayered γ-type tin phosphate, Sn(HPO_4_)_2_·3H_2_O, have revealed that this solid could be obtained in a water-delaminated form, which showed a high proton conductivity of ~1 × 10^−2^ S·cm^−1^ at 100 °C and 95% RH (Table 1). This high value was attributed to strong H-bonds between water and the SnP layers, in combination with a high surface area of 223 m^2^·g^−1^ [88].

### 2.4. Other Tetravalent Phosphates

POH groups-containing silicophosphates have received attention as proton-conducting electrolytes [42] but its solid-state structural characterization is elusive, at least, under conditions similar to those used in operating fuel cells [89]. A conductivity value of 2.5 × 10^−3^ S·cm^−1^ at 180 °C, under more than 0.4% RH, was reported for a silicophosphate-based composite membrane. However, it was noted that the formation of Si_5_O(PO_4_)_6_ crystals, under dry conditions, was detrimental for the proton conductivity.

NASICON-type compounds with the general formula, A*_x_*B_2_(PO_4_)_3_, where A = H^+^, H_3_O^+^, or NH_4_^+^ and B = Zr, Ti, Hf, Sn, Ge, etc., are also potential proton conductors [90,91]. HZr_2_(PO_4_)_3_ was firstly prepared by Clearfield et al. [92] and Komorowski et al. [93] described the synthesis of a hydrogen form from Na_1 + *x*_Zr_2_Si*_x_*P_3–*x*_O_12_ by replacing Na^+^ ions with hydronium ions through ion exchange. This solid showed a bulk proton conductivity of 5.0 × 10^−4^ S·cm^−1^ at 100% RH and 25 °C, the proton transport mechanism being a Grotthuss-type one [93].

Other materials, such as HZr_2_(PO_4_)_3_·*n*H_2_O, NH_4_Zr_2_(PO_4_)_3_, or H_1 ± *x*_Zr_2 *x*_M*x*(PO_4_)_3_ · H_2_O (M = Nb, Y) [94,95], have been reported. Some NASICON-type compounds based on zirconium or titanium hydrogen phosphate exhibit remarkable proton conductivity while showing a high thermal stability, at least for the case of zirconium derivatives [96,97]. Less studied are the NASICON-type hafnium hydrogen phosphates. Among them, (NH_4_)_0.4_H_0.6_Hf_2_(PO_4_)_3_, obtained by thermal decomposition of the cubic NH_4_Hf_2_(PO_4_)_3_ at 550 °C, showed a proton conductivity of 1.2 × 10^−6^ S·cm^−1^ at 500 °C [90].

### 2.5. Tetravalent Pyrophosphates

In addition to acid metal(IV) phosphates, tetravalent metal pyrophosphates (MP_2_O_7_; M = Sn, Ce, Ti, Zr), in particular tin(IV) derivatives, have received considerable attention as proton conductors for intermediate temperature electrolyte applications. These solids combine high thermal stability (T ≥ 400 °C) with conductivities of 10^−3^–10^−2^ S·cm^−1^ in the temperature range of 100–300 °C, under anhydrous or low humidity conditions [98,99,100,101,102]. The origin of this proton conductivity was initially explained because of protons being incorporated into the framework, formed by metal(IV) octahedra and corner-sharing phosphate tetrahedral, after interaction with water vapor.

Defect sites, such as electron holes [103] and oxygen vacancies are believed to favour proton generation [104]. These protons may occupy hydrogen-bonding interstitial sites on either the M−O−P or the P−O−P bonds, thus, generating a hopping proton transport between these sites [105,106]. In support of this, ^1^H NMR studies revealed two signals corresponding to hydrogen-bonded interstitial protons at phosphate tetrahedral and metal octahedral sites in In^3+^-doped SnP_2_O_7_. Furthermore, indium doping seemed not to affect the proton conduction occurring by a hopping mechanism between octahedral and tetrahedral sites.

Thus, the increased proton conductivity is instead attributable to increased proton concentration [107]. The strategy of dopant insertion into the M(IV) pyrophosphate has been further extended to include other dopant divalent and trivalent metal ions, such as Mg^2+^, Al^3+^, Sb^3+^, Sc^3+^, and Ga^3+^, with a maximum conductivity of 0.195 S·cm^−1^ (Table 1) being reported for In_0.1_Sn_0.9_P_2_O_7_ [108]. However, recent studies on metal doped and undoped SnP_2_O_7_ suggested that it is in co-precipitated phosphorous-rich amorphous phases where the proton conductivity mostly resides, while the crystalline phase exhibit a very low conductivity (~10^−8^ S·cm^−1^ at 150−300 °C) [109].

Notwithstanding, the proton conductivity of the high temperature-sintered materials could be recovered by phosphoric acid treatment [110] or by carefully choosing the phosphate precursor. For the latter case, tetrabutylammonium phosphate-based SnP_2_O_7_/Nafion^®^ composite membranes showed an increased fuel cell performance, as compared to other analogous SnP_2_O_7_/Nafion^®^ composite membranes [111]. A major challenge in using metal(IV) pyrophosphates as electrolytes is forming dense pellets and the associated low open circuit voltages (OCVs) and considerable gas crossover [108].

Several studies to test applicability of composite membranes for ITFCs have been conducted. For this purpose, dispersion of Sb_0.2_Sn_0.8_P_2_O_7_ nanoparticles into H_3_PO_4_-doped PBI membranes were shown to exhibit enhanced conductivities and greater power density relative to the pristine membranes in the single cell tests [112]. Similar claims were made for composite membranes based on Sn_0.95_Al_0.05_P_2_O_7_ [113]. Other approaches, such as using graphite oxide (GO)/SnP_2_O_7_ or biphosphate-containing SnP_2_O_7_/Nafion composite membranes have been accomplished.

The former showed improved proton conductivity (~7.6 × 10^−3^ S·cm^−1^) peak power density of 18 mW·cm^−2^ at 220 °C and reduced fuel crossover [114]. The latter, aimed at increasing stability by establishing strong ammonium cation-biphosphate ion interactions, which prevents the loss of biphosphate and, hence, optimises the three-phase interface at the fuel cell electrodes. With these improvements, the H_2_/O_2_ fuel cell delivered a power density of 870 mW·cm^−2^ at 240 °C, with minimal performance loss even under exposure to gas containing 25% carbon monoxide [37].

Mg-doped cerium pyrophosphate, with optimal composition of Ce_0.9_Mg_0.1_P_2_O_7_, showed a remarkable performance as electrolyte into a fuel cell, generating electricity up to 122 mA·cm^−2^ at 0.33 V and displaying a peak power value of 40 mW·cm^−2^ when using 50% H_2_ at 240 °C. Prepared in disk form and sintered at 450 °C, the Mg-doped material exhibited a significantly high proton conductivity in the presence of moist air (P_H2O_ = 0.114 atm) between 160 and 280 °C. A maximum value of 4.0 × 10^−2^ S·cm^−1^ was found at 200 °C, (Table 1), which was even higher than that of CeP_2_O_7_ (3.0 × 10^−2^ S·cm^−1^, at 180 °C) [115].

Low crystallinity titanium pyrophosphates also exhibit high proton conductivity [116]. Values of 0.0019–0.0044 S·cm^−1^, at 100 °C and 100% RH were reported for these compounds. The proton conduction is attributable to the presence of H_2_PO_4_^−^ and HPO_4_^2−^ species, demonstrated by ^31^P MAS NMR and FTIR, and it occurs through a water-facilitated Grotthuss-type proton transport mechanism. Titanium phosphates are also prone to be functionalized with phosphonates groups. Thus, mixed titanium phosphate/phosphonates, Ti(HPO_4_)_1.00_(O_3_PC_6_H_4_SO_3_H)_0.85_(OH)_0.30_·*n*H_2_O [117], displayed an exceptional proton conductivity of 0.1 S·cm^−1^ at 100 °C (Table 1).

A structurally different proton-conductor metal pyrophosphate is the mixed template-containing vanadium nickel pyrophosphate, (C_6_H_14_N_2_)[NiV_2_O_6_H_8_(P_2_O_7_)_2_]·2H_2_O (Figure 6). Its crystal structure is composed of octahedrally coordinated V(IV) and Ni(II) interconnected through bridging pyrophosphate groups, thus creating a 3D framework of [NiV_2_O_6_H_8_(P_2_O_7_)_2_]^2−^ unit charged-compensated by protonated DABCO molecules. Hydrogen-bonding networks inside 3D channels facilitate the proton transport, and thus this material exhibits a remarkable proton conductivity of 2.0 × 10^−2^ S·cm^−1^ at 60 °C and 100% RH (Table 1) through a Grotthuss-type proton-transfer mechanism (E_a_ = 0.38 eV) [118].

## 3. Super Protonic Metal(I) Phosphates

Solid acid proton conductors, with stoichiometry M^I^HyXO_4_ (M^I^ = Cs, Rb; X = S, P, Se; y = 1, 2), have received much attention because they exhibit exceptional proton transport properties and can be used as electrolytes in fuel cells operated at intermediate temperatures (120−300 °C). The fundamental characteristics of these materials are the phase transition that occurs in response to heating, cooling or application of pressure [6,119], accompanied by an increase in proton conductivity of several orders of magnitude—referred to as *super protonic* conductivity (Figure 7).

This property has been associated with the delocalization of hydrogen bonds [120]. For CsH_2_PO_4_, a proton conductivity of 6 × 10^−2^ S·cm^−1^ (Table 1) was measured above 230 °C corresponding to the *super protonic* cubic (Pm−3m) phase [121] while it drastically drops in the low temperature phases. Recently, from an ab initio molecular dynamics simulation study of the solid acids CsHSeO_4_, CsHSO_4_ and CsH_2_PO_4_, it was concluded that efficient long-range proton transfer in the high temperature (HT) phases is enabled by the interplay of high proton-transfer rates and frequent anion reorientation.

In these compounds, proton conduction follows a Grotthuss mechanism with proton transfer being associated with structural reorientation [122]. The *super protonic* conductor CsH_2_PO_4_ is stable under humidified conditions (P_H2O_ = 0.4 atm) [6], but it dehydrates to CsPO_3_, via the transient phase Cs_2_H_2_P_2_O_7_, at 230−260 °C, according to the relationship log(P_H2O_/atm) = 6.11(±0.82) − 3.63(±0.42) × 1000/(T_dehy_/K) [123].

Several studies [6,124,125] have shown that CsH_2_PO_4_ can be employed as the electrolyte in fuel cells with good long-term stability. Thus, a continuous, stable power generation for both H_2_/O_2_ and direct methanol fuel cells operated at ~240 °C was demonstrated for this electrolyte when stabilized with water partial pressures of ~0.3−0.4 atm. In fact, high performances, corresponding to single cell peak power densities of 415 mW·cm^−2^, was achieved for a humidified H_2_/O_2_ system provided with a CsH_2_PO_4_ electrolyte membrane of only 25 µm in thickness [126].

Various strategies, including salt modification by cation and anion substitution [127,128,129,130], and mixing with oxide materials [131,132,133,134,135,136,137] or organic additives [121,138,139,140], have been explored to improve the solid acid performance. The preparation of these derivatives and composites involved different synthesis techniques, such as sol-gel, impregnation, thin-casting and electrospinning, depending on the state of the precursor materials and the desired product [6].

Studies on the effects of incorporating different substituents in the structure of CsH_2_PO_4_ have shown mixed results. Thus, while incorporation of Rb^+^, up to 19%, led to a maximum conductivity of 0.03 S·cm^−1^, at 240 °C, Ba^2+^-doping only increased the conductivity of the low temperature phase in two orders of magnitude [127]. On the other hand, substitution of phosphorus with Mo or W hardly modified the proton conductivity, as compared to the parent compound. For S-doping, a shift to lower temperatures of the phase transition was observed at low S-contents, while the conductivity increased only for the low temperature phase.

Mixing CsH_2_PO_4_ with H_3_PO_4_ or CsH_5_(PO_4_)_2_ resulted in composites Cs_1−x_H_2+x_PO_4_ (x_mole_= 0.01−0.05) with excesses of protons. In comparison with CsH_2_PO_4_, these composites exhibited super protonic phase transition at lower temperature (up to ~200 °C) and increased lower-temperature conductivities (σ_150 °C_~ 2 × 10^−3^ S·cm^−1^, x = 0.1) [141]. Addition of Cs_2_HPO_4_·2H_2_O to CsH_2_PO_4_ (up to an x_mole_ = 0.03) also resulted in an enhanced low-temperature conductivity and decreased the activation energy to E_a_ = 0.56 eV. For all these composites no significant changes in the high-temperature conductivity were observed, although the super protonic phase transition became more diffuse and almost disappeared in some case [141,142].

Improvements in the mechanical and proton conductivity properties, as well as in thermal stability of CsH_2_PO_4_-based electrolytes have been addressed by mixing with oxide materials, such as zirconia, silica, alumina, and titania (Table 1). Thus, the (1-x)Cs_3_(HSO_4_)_2_(H_2_PO_4_)/xSiO_2_ composite with x = 0.7 increase the proton conductivity up to 10^−2^ S·cm^−1^ in the range 60–200 °C. Moreover, the introduction of fine-particle silica reduced the jump in conductivity at the phase-transition temperature and shifted it to lower temperatures. Higher silica contents led to a decrease in the conductivity due to the disruption of conduction paths [132].

The use of acid-modified silica confers high thermal stability at low H_2_O partial pressure while maintaining high proton conductivity (10^−3^–10^−2^ S·cm^−1^, at 130–250 °C) [134]. A similar trend was found for the composites (1−x)CsH_2_PO_4_/xTiO_2_ and (1−x)CsH_2_PO_4_/xZrO_2_ (contents x = 0.1 and 0.2) [135,136]. High performance was also reported for the composite 8:1:1 CsH_2_PO_4_/NaH_2_PO_4_/ZrO_2_, with a stable conductivity of 2.23 × 10^−2^ S·cm^−1^ for 42 h being measured for the high temperature phase [137]. Nanodiamonds (ND) are another type of heterogeneous additive giving rise to the (1−x)CsH_2_PO_4_-xND (x = 0–0.5) composites exhibiting enhanced low temperature proton conductivities, while maintaining almost unaltered that of high temperature [140].

CsH_2_PO_4_ composites based on organic additives have been also intensively investigated. For example, binary mixtures containing N-heterocycles (1,2,4-triazole, benzimidazole and imidazole) displayed enhanced proton conductivity at temperatures below the super protonic phase transition (2 − 8 × 10^−4^ S·cm^−1^ at 174–190 °C) [138]. Another approach was reported in which the solid acid CsH_2_PO_4_ was combined with fluoroelastomer p(VDF/HFP), producing high conductive composite membranes (1−x)CsH_2_PO_4_-xp(VDF/HPF) (x = 0.05–0.25 wt%) with improved mechanical and hydrophobic properties, along with flexibility and reduced thickness.

However, a high concentration of p(VDF/HFP) rendered membranes with a reduced proton conductivity for the HT phase [141]. By using the polymer butvar (polyvinyl butyral) [121], composite membranes (1−x)CsH_2_PO_4_-xButvar (x < 0.2 wt%) were obtained showing, in a wide range of composition, a proton conduction behaviour analogous to the pure salt in the high temperature region but with increased low temperature conductivity by three orders of magnitude at x = 0.2.
materials-15-01292-t001_Table 1Table 1Proton conductivity data for selected monovalent and tetravalent metal phosphates and pyrophosphates.Compounds/DimensionalityTemperature (°C)/RH(%)Conductivity (S·cm^−1^)Ea (eV)Ref.***Tetravalent Metal Phosphates***



ZrP·0.8PrNH_2_·5H_2_O/*2D*20/901.2 × 10^−3^1.04[50]Zr(P_2_O_7_)_0.81_(O_3_POH)_0.38_/*2D*20/901.3 × 10^−3^0.19[51]Zr(O_3_POH)_0.65_(O_3_PC_6_H_4_SO_3_H)_1.35_/*2D*100/907.0 × 10^−2^---[52](NH_4_)_2_[ZrF_2_(HPO_4_)_2_]/3*D*90/951.45 × 10^−2^0.19[53](NH_4_)_5_[Zr_3_(OH)_3_F_6_(PO_4_)_2_(HPO_4_)]/*3D*60/984.41 × 10^−2^0.33[54](NH_4_)_3_Zr(H_2/3_PO_4_)_3_/*1D*90/951.21 × 10^−2^0.30[55]Ti_2_(HPO_4_)_4_/*1D*20/951.2 × 10^−3^0.13[83]Ti_2_O(PO_4_)_2_·2H_2_O (π-TiP)/*3D*90/951.3 × 10^−3^0.23[86]Ti(HPO_4_)_1_(O_3_PC_6_H_4_SO_3_H)_0.85_(OH)_0.30_·*n*H_2_O/*2D*100/--0.10.18[117]Sn(HPO_4_)_2_·3H_2_O/*2D*100/951.0 × 10^−2^---[88]α-ZrP_2_O_7_/*3D*300 1.0 × 10^−4^---[49](NH_4_)_3_Zr(H_2/3_PO_4_)_3_/*1D*180 1.45 × 10^−3^0.26[55]***Tetravalent Pyrophosphates***



TiP_2_O_7_/*3D*100/1004.4 × 10^−3^0.14[116](C_6_H_14_N_2_)[NiV_2_O_6_H_8_(P_2_O_7_)_2_]·2H_2_O/*3D*60/1002.0 × 10^−2^0.38[118]In_0.1_Sn_0.9_P_2_O_7_/*3D*3000.195---[108]Ce_0.9_Mg_0.1_P_2_O_7_/*3D*200 4.0 × 10^−2^---[115]CeP_2_O_7_/*3D*180 3.0 × 10^−2^---[115]***Super protonic Cesium Phosphates***



CsH_2_PO_4_/*3D*>230 6.0 × 10^−2^---[120]Cs_1-x_Rb_x_H_2_PO_4_/*3D*240 3.0 × 10^−2^0.92[127]Cs_1−x_H_2+x_PO_4_/*3D*150 2.0 × 10^−2^0.70[141](1−x)Cs_3_(HSO_4_)_2_(HPO_4_)/xSiO_2_200 1.0 × 10^−2^---[132](1-x)CsH_2_PO_4_/xTiO_2_230 2.0 × 10^−2^---[135](1-x)CsH_2_PO_4_/xZrO_2_250 2.6 × 10^−2^---[136]CsH_2_PO_4_/NaH_2_PO_4_/ZrO_2_230 2.23 × 10^−2^---[137]

## 4. Divalent and Trivalent Metal Phosphates

### 4.1. Divalent Metal Phosphates

Divalent transition metal phosphates show a great structural versatility, from 1D polymeric topologies through layered framework to 3D open-framework structures. Most of these solids are synthetized in the presence of organic molecules, which are retained as protonated guest species (amines, iminazole derivatives, etc.), thus, compensating the anionic charge of the inorganic framework. This is formed by the metal ion, mainly in octahedral or tetrahedral coordination environments, linked to the phosphate groups with different protonated degrees (H_x_PO_4_). The presence of the latter makes possible the formation of effective and extensive hydrogen bond networks with participation of water molecules. In addition, protonated guest species and water itself can act as proton carriers, thus, boosting proton conduction (Table 2) [12,13,14].

Several 3D open-framework M(II) phosphates have been reported [14,143], which consist of [CoPO_4_]_∞_^−^ or [Zn_2_(HPO_4_)_2_(H_2_PO_4_)_2_]^2−^ anionic frameworks that contain organic charge-compensating ions in their internal cavities. (C_2_N_2_H_10_)_0.5_CoPO_4_ exhibited negligible conductivity in anhydrous conditions; however, it displayed a relatively high water-mediated proton conductivity 2.05 × 10^−3^ S·cm^−1^ at 56 °C and 98% RH. On the other hand, the solid NMe_4_·Zn[HPO_4_][H_2_PO_4_] experiences a structural transformation from monoclinic (α) to orthorhombic (β) upon heating at 149 °C. Both polymorphs contain 12-membered rings composed of tetrahedral Zn^2+^ ions linked to protonated phosphate groups without changing Zn-O-P connectivity (Figure 8).

The α phase transforms into the β phase at high humidity and temperatures above 60 °C, and then reaches a proton conductivity of 1.30 × 10^−2^ S·cm^−1^ at 98% RH, a behaviour that might be attributed to the participation of adsorbed water molecules in creating H-bonding networks with effective pathways for proton conduction. The conductivity drastically decreases at 65 °C. In anhydrous conditions, the α phase exhibited a proton conductivity of ~10^−4^ S·cm^−1^ at 160 °C, similar values were found for other reported zinc phosphates at temperatures between 130 and 190 °C [12,144,145].

Membrane-electrode assembly prepared with the pelletized solid, gave an observed open circuit voltage (OCV) of 0.92 V at 190 °C measured in a H_2_/air cell, suggesting that the dominant conductive species are protons, and the proton transport from anode to cathode takes place through H-bond networks in the pellet.

As an example of 2D metal phosphate water-assisted proton conductors, (C_2_H_10_N_2_) [Mn_2_(HPO_4_)_3_](H_2_O), displayed a proton conductivity of 1.64 × 10^−3^ S·cm^−1^ under 99% RH at 20 °C. This proton conductivity was attributed to the formation of dense H-bond networks in the lattice, composed of Mn_3_O_13_ units-containing anionic layers [146], which provide efficient proton-transfer pathways for a Grotthuss-type proton transport at high RH.

Another way of improving the proton conductivity in layered divalent metal phosphates is favouring the formation of hydrogen bond networks by ion exchange. For instance [17], partial exchange of Na^+^ for ethylendiammonium yielded two new crystalline phases, with composition (C_2_H_10_N_2_)_x_Na_1−x_[Mn_2_(PO_4_)_2_] (x = 0.37 y 0.54), which influenced formation of extended hydrogen bond networks and concomitant increase in proton conductivity, from 2.22 × 10^−5^ S·cm^−1^ for the as-synthesized material (in ethylendiammonium form) to 1.3 × 10^−2^ S·cm^−1^ (x = 0.37) and 2.1 × 10^−2^ S·cm^−1^ (x = 0.54) at 99% RH and 30 °C.

This strategy of enhancing proton conductivity was further extended successfully to other 2D manganese phosphates [18]. A proton conductivity value as high as 7.72 × 10^−2^ S·cm^−1^ was reached at 30 °C and 99% RH for K^+^-exchanged compounds, which compares well with those of MOF-based open-framework materials [147,148,149]. The 1D solid [Zn_3_(H_2_PO_4_)_6_(H_2_O)_3_](Hbim) (Hbim= benzimidazole) prepared by mechanochemical synthesis is characterized by presenting a dual-function as proton conductor.

It loses the coordinated water by heating transforming into [Zn_3_(HPO_4_)_6_](Hbim), which exhibits an intrinsic proton conductivity higher than the hydrated form, reaching a value of 1.3 × 10^−3^ S·cm^−1^ at 120 °C, believed to be due to a rearrangement of the conduction path and the liquid-like behaviour of benzimidazole molecules. In addition, this solid also showed porosity, thus, enabling the adsorption of gaseous methanol that further improved the proton conductivity of the anhydrous phase. This enhancement of proton conductivity in methanol-adsorbed samples was explained by the effective participation of the guest molecule in formation of extended hydrogen-bond interactions [19].

An ordered-to-disordered structural transformation and its implication in proton conduction were investigated for the 1D copper phosphate [ImH_2_][Cu(H_2_PO_4_)_2_Cl]·H_2_O (Im = imidazole). In this structure, the protonated imidazole (ImH_2_) and the water molecule are located in interspaces of the anionic chains [Cu(H_2_PO_4_)_2_Cl]^−^. Upon heating a structural transformation from an ordered crystalline state to a disordered state occurred. Highly mobile and structurally disordered H^+^ carriers were supposed to be responsible of the high proton conductivity 2 × 10^−2^ S·cm^−1^ at 130 °C, under anhydrous conditions [150].

A 1D zinc phosphate-based proton conductor, [Zn_3_(H_2_PO_4_)_6_(H_2_O)_3_](BTA) (BTA = 1,2,3-benzotriazole) has been reported [151] that exhibits high proton conductivity, 8 × 10^−3^ S·cm^−1^ in anhydrous glassy-state (120 °C). The glassy-state, developed via melt-quenching, was suggested to induce isotropic disordered domains that enhanced H^+^ dynamics and conductive interfaces. In fact, the capability of the glassy-state material as an electrolyte was found suitable for the rechargeable all-solid-state H^+^ battery operated in a wide range of temperatures from 25 to 110 °C.

Focus has been also put on metal phosphate-based solid solutions [152,153]. Vacancies can be generated that introduce extra protons into the structure and increase the proton conduction. This effect was investigated for the 1D rubidium and magnesium polyphosphate compound, RbMg_1-x_H_2x_(PO_3_)_3_·yH_2_O, for which system a maximum proton conductivity of 5.5 × 10^−3^ S·cm^−1^ was measured at 170 °C with a vehicle-type mechanism of H_3_O^+^ conduction.

The proton conductivity results from H-bond interactions between water molecules and corner-sharing PO_4_ chains that provide formation of sandwiched edge-sharing RbO_6_-MgO_6_ chain [152]. Another example is the solid solution with composition Co_1-x_Zn_x_(H_2_PO_4_)_2_·2H_2_O (0 < x < 1.0), which showed the highest conductivity value, 2.01 × 10^−2^ S·cm^−1^ at 140 °C, for a composition Co_0.5_Zn_0.5_(H_2_PO_4_)_2_·2H_2_O [153].

### 4.2. Trivalent Metal Phosphates

Zeolite-like open framework metal(III) phosphates consist of metal(III) ions-phosphate species (PO_4_^3-^, HPO_4_^2-^, H_2_PO_4_^−^) linkages featuring internal cavities, where charge-compensating cations and/or neutral species are located and, thus, dense hydrogen bond networks frequently result. In addition, the robust inorganic framework endues this porous material with better thermal and chemical stability compared with porous coordination polymers/metal organic frameworks (PCPs/MOFs) [13].

Regarding to proton conductivity (Table 2), aluminium phosphate-based solids are by far the most studied compounds [154,155,156,157,158,159,160,161,162,163,164,165,166,167]. The species inside channels affect in different ways to proton conduction. Thus, while water adsorption is key to assist proton transfer in (NH_4_)_2_Al_4_(PO_4_)_4_(HPO_4_)·H_2_O by a hopping mechanism along H-bond chains [166], densely packed NH_4_^+^ ions show negligible contribution because of hampered migration.

By using an organic template-free synthetic methodology, a 3D open-framework aluminophosphate Na_6_[(AlPO_4_)_8_(OH)_6_]·8H_2_O (JU103) was prepared [158], which showed a proton conductivity of 3.59 × 10^−3^ S·cm^−1^, at 20 °C and 98% RH. It was argued that the enhanced conductivity of the as-synthesized material as compared to its NH_4_^+^- or Ag^+^-exchanged forms is indicative of a beneficial effect of hydrated Na^+^ ions in generating proton-transfer pathways. The 3D cesium silicoaluminophosphate, Cs_2_(Al_0.875_Si_0.125_)_4_(P_0.875_Si_0.125_O_4_)_4_(HPO_4_), belonging to the structural family of SAPOs, was shown to exhibit a remarkable proton conductivity of 1.70 × 10^−4^ S·cm^−1^ at low temperature and RH (20 °C and 30%, respectively) [168,169].

Several 2D aluminophosphates have been reported as proton conductors [154,156]. These compounds (denoted as AlPO-CJ70/2) are structurally characterized by displaying an anionic layer: [Al_2_P_3_O_12_]^3-^, formed by alternating Al^3+^ and phosphorus tetrahedra, which is charge-compensated by N,N-dimethylbenzylamine or α-methylbenzylamine ions, respectively. In these layered structures, extended H-bond networks are formed through interactions of the amine N atoms, H_2_O molecules and protruding phosphate groups of the anionic layer. Consequently, water-mediated proton conduction processes occurred upon immersion in water, with σ values around 10^−3^ S·cm^−1^, at 80 °C, and E_a_ values of 0.16−0.2 eV, typical of a Grotthuss-type proton-transfer mechanism.

By following a synthetic route in which methylimidazolium dihydrogenphosphate was used as a solvent, structure-directing agent, and a phosphorus source, the solid (C_4_H_7_N_2_)(C_3_H_4_N_2_)_2_·Al_3_(PO_4_)_4_·0.5H_2_O (SCU−2) was obtained [164]. Its layered structure, built up from corner-sharing Al^3+^ and P^V^ tetrahedra, features 8-member rings where guest imidazolium ions and water molecules are hosted. At 85 °C and 98% RH, this solid showed a proton conductivity of 5.9 × 10^−3^ S·cm^−1^ and a low activation energy (0.20 eV), characteristic of a Grotthuss-type proton-transfer mechanism. An efficient pathway for the proton transfer was attributed to the hydrogen bond network established between the imidazolium ions and water molecules interacting with the host framework.

A few examples of phosphate-based proton conductors of other trivalent metals do exist. Among them, two Fe(III) phosphates, 1D (C_4_H_12_N_2_)_1.5_[Fe_2_(OH)(H_2_PO_4_)(HPO_4_)_2_(PO_4_)]·0.5H_2_O [13] and 3D open-framework iron(III) phosphate (NH_3_(CH_2_)_3_NH_3_)_2_[Fe_4_(OH)_3_(HPO_4_)_2_(PO_4_)_3_]·4H_2_O [170], have been reported. Both compounds contain Fe_4_O_20_ tetramers as a common structural feature. The 1D solid is composed of chains of tetramers bridged by PO_4_^3-^ groups and having terminal H_2_PO_4_^−^ and HPO_4_^2−^ groups [171], while piperazinium cations and water molecules are disorderly situated in between chains. This arrangement gives rise to extended hydrogen bonding interactions and hence proton conducting pathways. The proton conductivity measured at 40 °C and 99% RH was 5.14 × 10^−4^ S·cm^−1^, and it was maintained upon dispersion of this solid in PVDF [13]. In the case of the 3D solid, infinite chains of interconnected tetramers are interlinked, in turn by phosphate groups that generate large tunnels (Figure 9). The diprotonated 1,3-diaminopropane and water molecules, localized inside tunnels, form an extended hydrogen bond network with the P–OH groups pointing toward cavities. These interactions favour proton hopping, the measured proton conductivity being of 8.0 × 10^−4^ S·cm^−1^, at 44 °C and 99% RH, and with an E_a_ of 0.32 eV [172]. Furthermore, the proton conductivity of this compound increased up to 5 × 10^−2^ S·cm^−1^ at 40 °C upon exposure to aqua-ammonia vapors from 1 M NH_3_·H_2_O solution. This result confirms this treatment as an effective way of enhancing proton conductivity, which has been elsewhere demonstrated for the case of coordination polymers [173,174]. The observed variation of E_a_ with the ammonia concentration also suggested that NH_3_, as well as H_2_O, molecules contribute to create proton-transfer pathways, by a Grotthuss mechanism. However, when ammonia concentrations were lower than 0.5 M, the proton conduction mechanism tended to be vehicle-type one [156].
materials-15-01292-t002_Table 2Table 2Proton conductivity data for selected divalent and trivalent metal phosphates.Compounds/DimensionalityTemperature (°C)/RH(%)Conductivity (S·cm^−1^)Ea (eV)Ref.***Divalent Metal Phosphates***



(C_2_N_2_H_10_)_0.5_CoPO_4_/*3D*56/982.05 × 10^−3^1.01[143]NMe_4_·Zn[HPO_4_][H_2_PO_4_] (β phase)/*3D*60/981.30 × 10^−2^0.92[14](C_2_H_10_N_2_) [Mn_2_(HPO_4_)_3_](H_2_O)/*2D*20/991.64 × 10^−3^0.22[146](C_2_H_10_N_2_)_x_Na_1−x_[Mn_2_(PO_4_)_2_]/*2D*30/992.1 × 10^−2^0.14[17](C_2_H_10_N_2_)_1-x_K_x_[Mn_2_(PO_4_)_2_]·2H_2_O/*2D*30/997.72 × 10^−2^0.18[18](C_2_H_10_N_2_)_1-x_K_x_[Mn_2_(HPO_4_)_3_] (H_2_O)/*2D*30/990.85 × 10^−2^0.081[18][Zn_3_(H_2_PO_4_)_6_(H_2_O)_3_](Hbim)/*1D*120 1.3 × 10^−3^0.50[19][ImH_2_][Cu(H_2_PO_4_)_2_Cl]·H_2_O/*1D*130 2.0 × 10^−2^0.1[150][Zn_3_(H_2_PO_4_)_6_(H_2_O)_3_](BTA)*/1D*120 8.0 × 10^−3^0.39[151]RbMg_0.9_H_0.2_(PO_3_)_3_·*y*H_2_O/*1D*170 5.5 × 10^−3^---[152]Co_0.5_Zn_0.5_(H_2_PO_4_)_2_·2H_2_O/*1D*140 2.01 × 10^−2^---[153]***Trivalent Metal Phosphates***



Na_6_[(AlPO_4_)_8_(OH)_6_]·8H_2_O/*3D*20/983.59 × 10^−3^0.21[158][C_9_H_14_N]_8_[H_2_O]_4_·[Al_8_P_12_O_48_H_4_]/*2D*80, in water9.25 × 10^−4^0.16[154][*R*-,S-C_8_H_12_N]_8_[H_2_O]_2_·[Al_8_P_12_O_48_H_4_]/*2D*90/983.01 × 10^−3^0.20[156](C_4_H_7_N_2_)(C_3_H_4_N_2_)_2_·Al_3_(PO_4_)_4_·0.5H_2_O/*2D*85/985.94 × 10^−3^0.20[164]In(HPO_4_)(H_2_PO_4_)(D,L-C_3_H_7_NO_2_)/*3D*85/982.9 × 10^−3^0.19[16](NH_3_(CH_2_)_3_NH_3_)_2_[Fe_4_(OH)_3_(HPO_4_)_2_(PO_4_)_3_]·4H_2_O/*1D*40/995.0 × 10^−2^---[170]Hbim = benzimidazole; Im =imidazole; BTA = 1,2,3-benzotriazole.

For the series of isostructural imidazole cation (ImH_2_)-templated layered metal phosphates, [ImH_2_][X-(HPO_4_)_2_(H_2_O)_2_] (FJU−25-X, X = Al, Ga, and Fe), it was found that the proton conductivity was dependent on mobility of imidazole guests, FJU−25-Fe exhibiting the highest proton conductivity (5.21 × 10^−4^ S·cm^−1^ at 90 °C). The determined activation energies (~0.20 eV) were indicative of a Grotthuss-type mechanism of proton conduction.

The amino acid-template indium phosphate, In(HPO_4_)(H_2_PO_4_)(D,L-C_3_H_7_NO_2_) (SCU−12), represents a singular example of 3D metal phosphate-based proton conductors [16].

Its crystal structure is formed by edge-sharing four-ring ladders, with the amino acid molecules attached to the ladders through In–O bonds. Further bridging the indium phosphate ladders by the H_2_PO_4_^−^ groups gives rise to a three-dimensional structure. The presence of two kinds of proton carriers, H_2_PO_4_^−^ ions and zwitterionic alanine molecules, favours the development of high proton conductivities (2.9 × 10^−3^ S·cm^−1^ at 85 °C and 98% RH) through a Grotthuss-type proton-transfer mechanism (E_a_ = 0.19 eV).

Other trivalent metal phosphates have been reported, e.g., BPO_x_ [175] and CePO_4_ [176]. The former exhibited a proton conductivity of 7.9 × 10^−2^ S·cm^−1^ as self-supported electrolyte and 4.5 × 10^−2^ S·cm^−1^ as (PBI)−4BPO_x_ composite membrane, measured at 150 °C and 5% RH, but structure/conductivity correlations were not established because of its amorphous nature. The latter showed a low-temperature (RT) proton conduction < 10^−5^ S·cm^−1^ at 100% RH through a structure-independent proton-transport mechanism [176].

There are only a few examples of metal(III) pyrophosphates displaying proton conductivity. Among them is the open framework magnesium aluminophosphate MgAlP_2_O_7_(OH)(H_2_O)_2_ (JU102) [167]. Its structure is composed of tetrahedral Al^3+^ and octahedral Mg^2+^ ions coordinated by pyrophosphate ions. This connectivity results in an open framework with unidirectional 8-ring channels. The proton conduction properties originate from the existence of an H-bond network in which coordinated water molecules participate.

Thus, the proton conductivity measured at 55 °C on water-immersed samples was 3.86 × 10^−4^ S·cm^−1^ [167], which raised to 1.19 × 10^−3^ S·cm^−1^ when calcined at 250 °C and measured at the same conditions, while the E_a_ value hardly changed from 0.16 to 0.2 eV. This behaviour was explained as being due to a dehydration–rehydration process that enhances proton conductivity by altering the H-bonding network and the pathway of proton transfer.

Another example of 3D open framework metal(III) pyrophosphate is the compound NH_4_TiP_2_O_7_ [177]. The structure of this solid is composed of negatively charged [TiP_4_O_12_]^−^ layers, forming one-dimensional six-membered ring channels, where the NH_4_^+^ ions are located. Its proton conductivity increased from 10^−6^ S·cm^−1^ under anhydrous conditions to 10^−3^ S·cm^−1^ at full-hydration conditions and 84 °C. The low E_a_ value, 0.17 eV, characteristic of a Grotthuss-type proton-transfer mechanism, was associated with the role played by the NH_4_^+^ ions in the channels as proton donors and promoters of proton migration. A drop in proton conductivity was observed when the triclinic TiP_2_O_7_ phase formed by thermal decomposition of NH_4_TiP_2_O_7_ [177].

## 5. Outlook

Great efforts have been devoted to the synthesis and characterization of metal phosphate-based proton conductors over more than three decades. Among them, zirconium phosphates are prominent not only because of the feasibility of Zr(IV) and phosphate ions to form a rich variety of crystallographic architectures (from 1D to 3D open frameworks) but also due to their outstanding properties and workability while being environmentally benign and low cost materials.

Although the prototype layered α-zirconium phosphate has been commonly proven as a filler for PEMFCs devices, new synthetic designs of M(IV) phosphates, including pyrophosphate compounds, are promising candidates to broaden their applicability in different electrochemical devices. Other M(IV) phosphates and pyrophosphates (M = Ti, Sn) are less known due, in part, to their amorphous nature or strong tendency to amorphise at working temperatures, though, in some cases, these compounds presented remarkable proton conductivity properties.

CsH_2_PO_4_, a super protonic material, was proven as a suitable electrolyte for both H_2_/O_2_ and direct methanol fuel cells operated at ~240 °C, and provides excellent performance when controlling its thermal stability. In addition, combinations with other materials are possible to adjust the specific characteristics of the composite CsH_2_PO_4_ electrolyte, thus, offering a wide range of compositions with tuning properties.

Recently, research and developments in metal phosphate proton conductors have been addressed to divalent or trivalent metal phosphates, which present a remarkable structural versatility and tunable conductivities; however, more in-depth studies are required to assess their potential use and applicability for low and intermediate temperature fuel cells.

Applications of metal phosphates as electrolytes or as electrolyte components are in continuous progress, although their use for energy storage and conversion remains a challenge. For practical applications in fuel cells at low/intermediate temperatures, phosphate-based proton conducting electrolytes have demonstrated acceptable proton conductivity values; however, other features, such as their mechanical strength, chemical/thermal stabilities, film-forming ability (in the case of composite membranes), durability, and fuel cross-over, are key factors to be improved.

## Figures and Tables

**Figure 1 materials-15-01292-f001:**
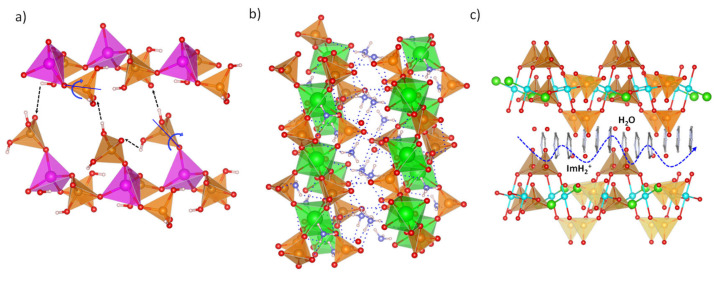
Examples of proton transport in phosphate-containing materials: (**a**) structural reorientation-mediated proton transfer. Zn (magenta), P (orange), O (red) and H (pale pink). (**b**) hopping through H-bond networks, and (**c**) carrier-mediated proton conduction. Zn (magenta), Zr (green), Cu (cyan), Cl (light green), P (orange), O (red), N (blue) and H (pale pink) atoms.

**Figure 2 materials-15-01292-f002:**
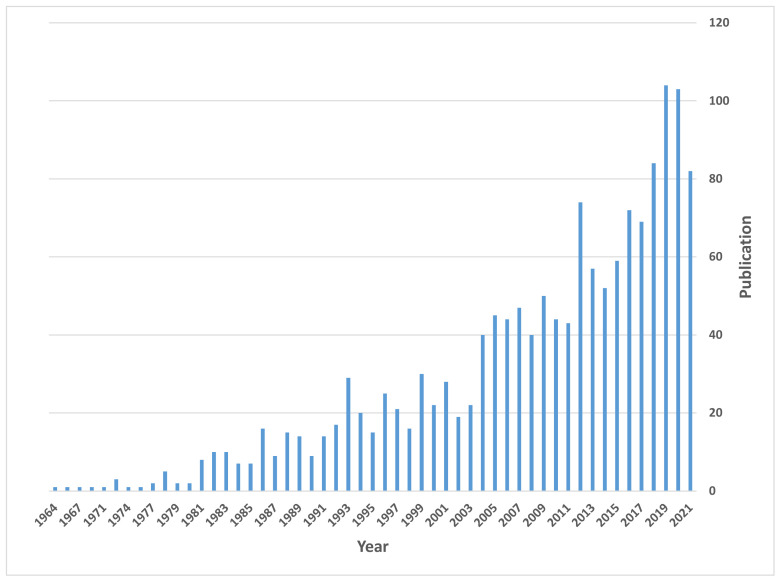
Publications on metal phosphates and pyrophosphate proton conductors over time.

**Figure 3 materials-15-01292-f003:**
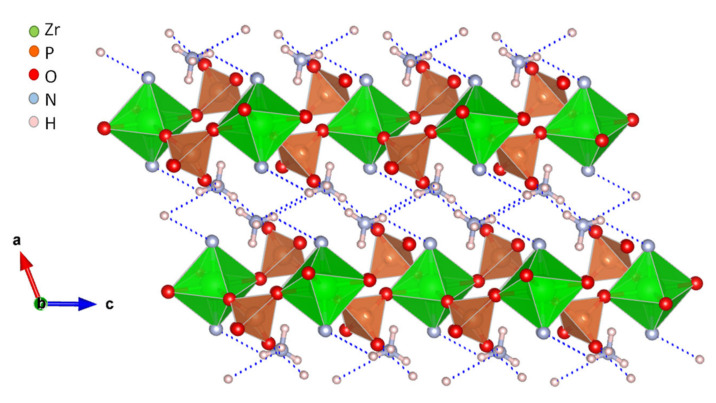
Layered structure of compound (NH_4_)_2_[ZrF_2_(HPO_4_)_2_] and possible proton-transfer pathways (adapted from [53]).

**Figure 4 materials-15-01292-f004:**
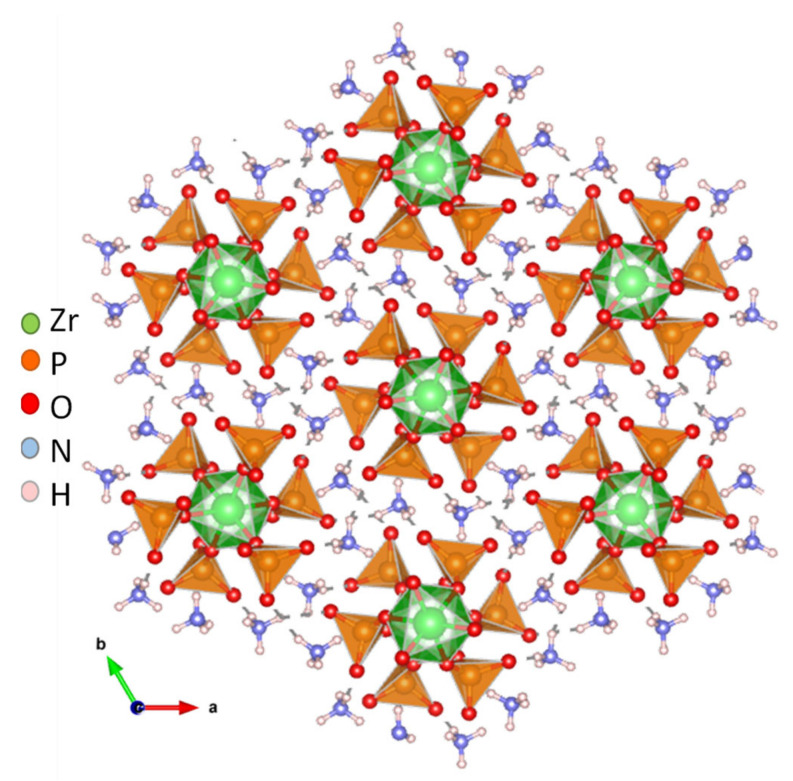
Views of chain packing and H-bond network (dashed grey lines) in compound (NH_4_)_3_Zr(H_2/3_PO_4_)_3_ (adapted from [55]).

**Figure 5 materials-15-01292-f005:**
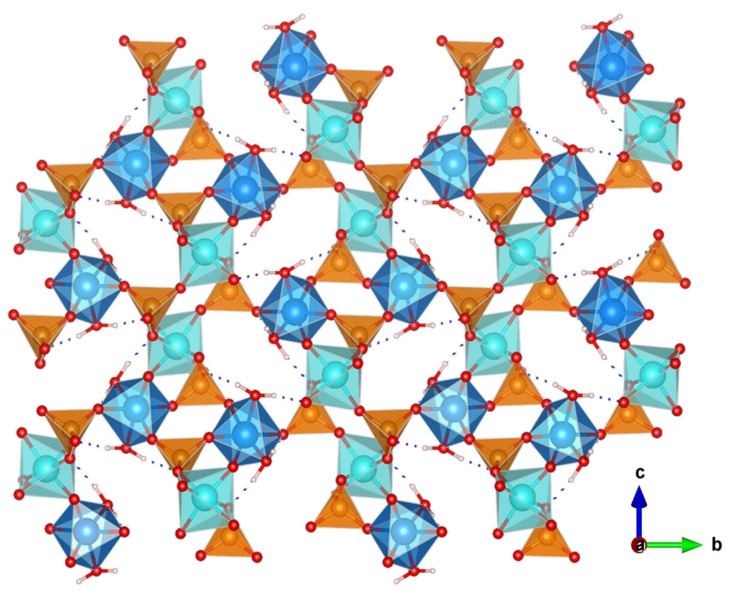
Structure of Ti_2_O(PO_4_)_2_·2H_2_O, a fibrous compound with capability of enhancing surface proton transport by H_3_PO_4_ impregnation (adapted from [86]).

**Figure 6 materials-15-01292-f006:**
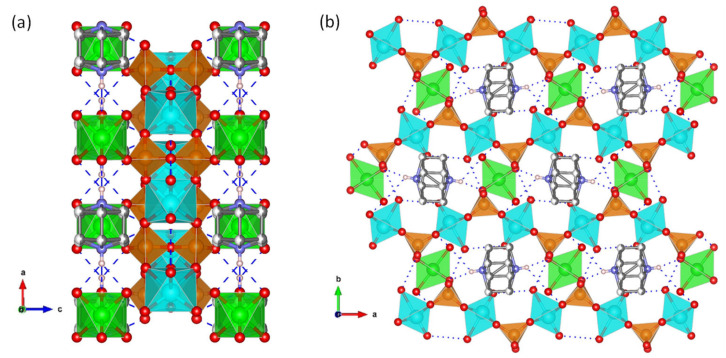
(**a**) View, along *b-axis*, of the chain packing and (**b**) 3D-framework of compound (NH_4_)_3_Zr(H_2/3_PO_4_)_3_, with H-bond network (dashed blue lines) highlighted (adapted from [118]).

**Figure 7 materials-15-01292-f007:**
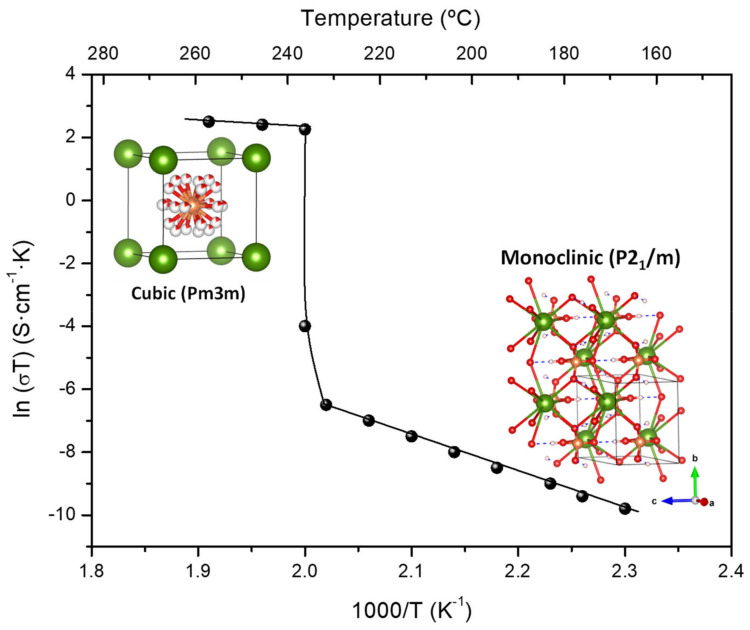
The characteristic high and low temperature proton conductivity and the corresponding structures for CsH_2_PO_4_, adapted from [6,120].

**Figure 8 materials-15-01292-f008:**
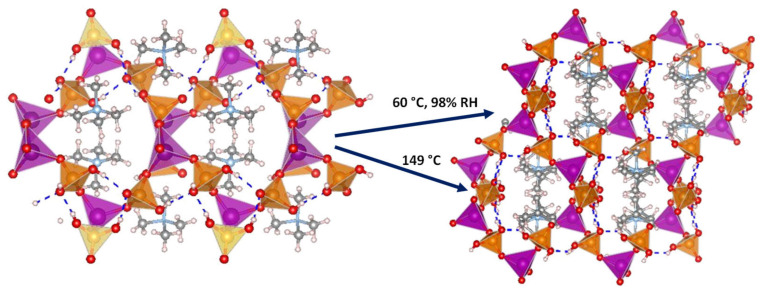
Irreversible structural transformation in NMe_4_Zn[HPO_4_][H_2_PO_4_]_4_ adapted from [143]. N (sky-blue), O (red), Zn (magenta), P (orange), C (grey) and H (pale pink) atoms.

**Figure 9 materials-15-01292-f009:**
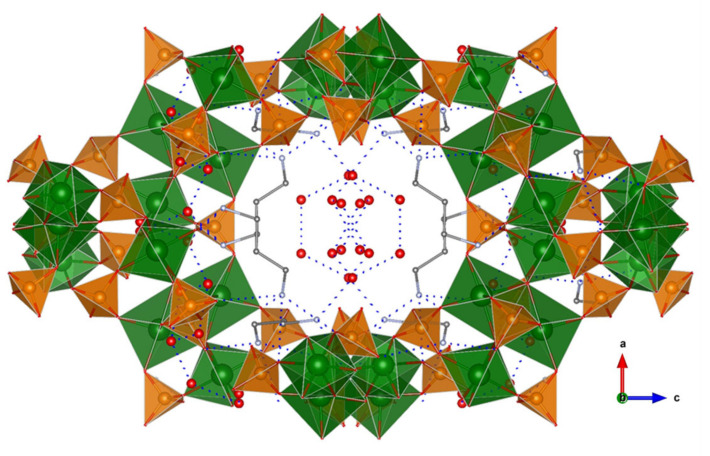
Open-framework structure of (NH_3_(CH_2_)_3_NH_3_)_2_[Fe_4_(OH)_3_(HPO_4_)_2_(PO_4_)_3_]·4H_2_O showing guest species inside channels and H-bond interactions. Fe (green), O (red), P (orange), and C (grey) atoms.

## Data Availability

Not applicable.

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
