# Peer review of "Properties and Applications of Metal Phosphates and Pyrophosphates as Proton Conductors"

_materials, 2022, doi:10.3390/ma15041292_

Round 1
Reviewer 1 Report
This manuscript made a review of the metal phosphates and focused on their proton conducting properties. This review was outlined mainly according to the metal oxidation states, which make the paper clear and easy to follow. At the end of this review, an outlook for the metal phosphates was made, with the most prominent zirconium phosphate being stressed, as well as the developing trend for the divalent and trivalent metal phosphates being proposed. This is a well written and organized review, and can provide a good reference for those who would like to do some research in metal phosphates. I suggest it to be published without revision.
Author Response
We thank the reviewer for the time and effort that he has put into reviewing the draft manuscript.
Reviewer 2 Report
I have reviewed an article entitled "Proton conductivity properties and applications of metal phosphates" written by Rosario M.P. Colodrero et. al. The article discussed on proton conductor of metal phosphates in terms of conductivity and application, structure properties relationship and main characteristic of super protonic C5H2PO4. Although the intensive elaboration was carried out by the authors, there are some info need to be included in the manuscript so that it will be more informative and increase the scientific value. The required info are as follow:
1) The trend of publication and patent for this topic
2) Schematic diagram to picture the article content
3) Some of the mechanisms in the manuscript should be in Figure form.
4) It is good if some of data in the manuscript are presented in Table.
5) Report or info on the metal phosphates as electrolyte in pilot scale production needs to be highlighted.
Author Response
The required info are as follow:
1) The trend of publication and patent for this topic.
Answer: Agreed. We have included a graphical trend of publications for this topic in the revised version of the manuscript.
2) Schematic diagram to picture the article content.
Answer: Agreed. We have included a schematic diagram as Figure 1.
3) Some of the mechanisms in the manuscript should be in Figure form.
Answer: Agreed. Figures are been included to illustrate some specific proton conduction mechanisms.
4) It is good if some of data in the manuscript are presented in Table.
Answer: Agreed. Data featuring solid properties are summarised in tables 1 and 2 in the revised manuscript.
5) Report or info on the metal phosphates as electrolyte in pilot scale production needs to be highlighted.
Answer: We have not found reports about pilot scale production of the studied materials.
Reviewer 3 Report
This is a good paper and will find interesting to many academics in the field.
I am not like the name Proton exchange membrane fuel cells (PEMFCs). I prefer Polymer Electrolyte Membrane fuel cell (PEMFC).
Nafion needs a TM - Nafion™ (This paper is correctly used.) However, there are a of different companies with Perfluorinated sulfonic acid (PFSA) and should discuss this also.
Most methanol PEMFC will operate at above 250 degrees C in higher pressure and this paper should be clear that this must be pressurized and what the degradation of the materials.
Some typos and the ‘English’ can be improved.
Overall – this is a good paper. HOWEVER – this journal paper is ‘unreadable’ because of the failure to have Figures, Photos (images), and Tables. There are required images of the chemical structures. There are opportunities of membrane and structure of water movement (or a schematic of the water structure movement). Tables of the different the ‘14’ materials (the paper is discussed this but there are no Tables). And more Sub-Headings.
Author Response
1) I am not like the name Proton exchange membrane fuel cells (PEMFCs). I prefer Polymer Electrolyte Membrane fuel cell (PEMFC).
Answer: Agreed. We have changed “Proton-exchange membrane fuel cells” and “proton exchange membranes”, by “Polymer electrolyte membrane fuel cells” and “polymer electrolyte membranes”, respectively.
2) Nafion needs a TM - Nafion™ (This paper is correctly used.) However, there are a of different companies with Perfluorinated sulfonic acid (PFSA) and should discuss this also.
Answer: Agreed. We have changed “perfluorosulfonated ionomers, such as Nafion®,” by “perfluorinated sulfonic acid (PFSA) ionomers, such as Nafion® and Aquivion®”. In addition, some results based on Aquivion PFSA were also discussed in the manuscript (page 4).
3) Most methanol PEMFC will operate at above 250 degrees C in higher pressure and this paper should be clear that this must be pressurized and what the degradation of the materials.
Answer: According with the suggestion of the reviewer we have included the following sentence in introduction of the revised manuscript:
Nevertheless, for the case of methanol HT-PEMFCs, that must operate above 250 °C and high pressure, possible degradation of the electrolytes should be a subject of study [40].
4) Some typos and the ‘English’ can be improved.
Overall – this is a good paper. HOWEVER – this journal paper is ‘unreadable’ because of the failure to have Figures, Photos (images), and Tables. There are required images of the chemical structures. There are opportunities of membrane and structure of water movement (or a schematic of the water structure movement). Tables of the different the ‘14’ materials (the paper is discussed this but there are no Tables). And more Sub-Headings.
Answer: According with the suggestion of the reviewer different figures relative to crystal structure of selected materials are given in the revised version of the manuscript. A Table summarizing proton conductivity data have been also included. In addition, some more sub-headings have been incorporated.
Reviewer 4 Report
Journal: "Materials"
Title: Proton conductivity properties and applications of metal phosphates.
Authors: Rosario M.P. Colodrero, Pascual Olivera-Pastor, Aurelio Cabeza and Montse Bazaga-García.
In this work, the authors summarized a review of proton conductive properties and applications of metal phosphates.
The authors should improve some recommendations. Such as:
1- There are many grammatical as well as English structural issues that need to be considered. Generally, the English language should be improved in some places.
In Abstract section: Line 13 & 22.
In Introduction section: Line 29, 32, 42, 46, 63, 65, 69, 77, 91, & 100.
In 2.1 section: Line 116, 127, 128, 172, 175, 184, 237, 241, 245, 260.
In 2.2 section: Line 293, 297, 315, 316.
In 3 section: Line 346, 357, 363, 366, 367, 391, 417.
In 4.1 section: Line 424, 451, 454, 481, 485, 491, 492.
In 4.2 section: Line 512, 520, 521, 559, 568, 594, 598.
In 5 section: Line 602, 612, 617, 625.
2- Line 344, what about M?
3- Line 136, How (which is ~300 times higher)? Line 351, this value is given not right this should be different according to given references. It should be revised.
4- Line 363, confusing about the same property.
5- Line 387 & 388, similar terms were not used for low temperature (LT).
6- Line 577-579, Rare earth metal phosphates should also be discussed in this revision. Their proton conductivity was sufficient for practical applications. It is also better to study the effect of high temperature on conductivity.
7- Line 589, How does the Ea value remain unchanged?
8- No attempt is made to describe the proton conductivity of metal phosphates under humid conditions. The authors should include proton conductivity data under humid conditions.
9- What is the mechanism of selecting the compound of these materials? Explain the novelty of your work on other studies.
10- The authors are encouraged to add a paragraph to the introduction concerning the applications of given materials. It must be added.
Author Response
1) There are many grammatical as well as English structural issues that need to be considered. Generally, the English language should be improved in some places.
Answer: We have revised throughout the manuscript all grammatical mistakes.
- In Abstract section: Line 13 & 22. Done
- In Introduction section: Line 29, 32, 42, 46, 63, 65, 69, 77, 91, & 100. Done
- In 2.1 section: Line 116, 127, 128, 172, 175, 184, 237, 241, 245, 260. Done
- In 2.2 section: Line 293, 297, 315, 316. Done
- In 3 section: Line 346, 357, 363, 366, 367, 391, 417. Done
- In 4.1 section: Line 424, 451, 454, 481, 485, 491, 492. Done
- In 4.2 section: Line 512, 520, 521, 559, 568, 594, 598. Done
- In 5 section: Line 602, 612, 617, 625. Done
2) Line 344, what about M?
Answer: The formula has been corrected as follows: MIHyXO4 (MI = Cs, Rb; X = S, P, Se; y = 1, 2)
3) Line 136, How (which is ~300 times higher)?
Answer: Agreed. We have changed “which is ~300 times higher than that measured in the perpendicular direction”, by “This conductivity was found to be ~2 order of magnitude higher than that measured on applying the electric field perpendicular to the pellicles [49].”
4) Line 351, this value is given not right this should be different according to given references. It should be revised.
Answer: Agreed. Reference is just [121].
5) Line 363, confusing about the same property.
Answer: Agreed. For clarity, we have rewritten the sentence “Despite of the apparently poor thermal stability of this material, a continuous, stable power generation for both H2/O2 and direct methanol fuel cells operated at ~240 °C was demonstrated for this electrolyte when stabilized with water partial pressures of ~0.3-0.4 atm.” as follows:
“So, a continuous, stable power generation for both H2/O2 and direct methanol fuel cells operated at ~240 °C was demonstrated for this electrolyte when stabilized with water partial pressures of ~0.3-0.4 atm."
6) Line 387 & 388, similar terms were not used for low temperature (LT).
Answer: Agreed. We have changed “LT” by “lower-temperature”
7) Line 577-579, Rare earth metal phosphates should also be discussed in this revision. Their proton conductivity was sufficient for practical applications. It is also better to study the effect of high temperature on conductivity.
Answer: Because high temperature proton conductors are out of the scope of this review, the following paragraph ¨Other trivalent metal phosphates (e.g. boron) [175] and rare earth metal phosphates have been also reported [176-178] but, to our knowledge, their proton conductivities were not sufficiently high for practical applications and, therefore, they will not be discussed further in this revision¨ has been rewritten as follows:
Other trivalent metal phosphates have been reported, e.g. BPOx [175] and CePO4 [176]. The former exhibited a proton conductivity of 7.9×10-2 S·cm-1 as self-supported electrolyte and 4.5×10-2 S·cm−1 as (PBI)-4BPOx composite membrane, measured at 150 °C and 5% RH, but structure/conductivity correlations were not established because of its amorphous nature. The latter showed a low-temperature (RT) proton conduction < 10-5 S·cm-1 at 100 % RH through a structure-independent proton-transport mechanism [176].
8) Line 589, How does the Ea value remain unchanged?
Answer: We have rewritten line 589 as follows "when calcined at 250 °C and measured at the same conditions, while the Ea value hardly changed, from 0.16 to 0.2 eV."
9) No attempt is made to describe the proton conductivity of metal phosphates under humid conditions. The authors should include proton conductivity data under humid conditions.
Answer: We have included throughout the manuscript proton conductivity data reported at different relative humidities, especially for those low-temperature proton conductors.
10) What is the mechanism of selecting the compound of these materials? Explain the novelty of your work on other studies.
Answer: In this manuscript, we have intended to give an overview of metal phosphates focusing on those that have shown attractive properties as proton conductors, particularly worthy of being applied as electrolytes. Also, we tried to rationalize properties with specific structural features when appropriate, which might have an impact in future designs of these materials of fuel cells metal phosphate-based electrolytes. We believe that these points have been indicated in the introduction section.
11) The authors are encouraged to add a paragraph to the introduction concerning the applications of given materials. It must be added.
Answer: For clarity, we have modified the title of the manuscript in the revised version as follows:
“Properties and applications of metal phosphates and pyrophosphates as proton conductors”
Round 2
Reviewer 3 Report
Even more graphics on the figures would still help.
Reviewer 4 Report
Journal: "Materials"
Title: Properties and applications of metal phosphates and pyro-phosphates as proton conductors.
Authors: Rosario M.P. Colodrero, Pascual Olivera-Pastor, Aurelio Cabeza and Montse Bazaga-García.
In this work, the authors summarized a review of proton conductive properties and applications of metal phosphates and pyro-phosphates.
The authors should improve some recommendations. Such as:
1- Now, I think the English language has been improved which is matched the quality of the materials journal.
2- Line 1004, should be revised, Because the temperature is not at 230 °C it should be higher than 230 °C.